# The Kitty Microbiome Project: Defining the Healthy Fecal “Core Microbiome” in Pet Domestic Cats

**DOI:** 10.3390/vetsci9110635

**Published:** 2022-11-16

**Authors:** Holly H. Ganz, Guillaume Jospin, Connie A. Rojas, Alex L. Martin, Katherine Dahlhausen, Dawn D. Kingsbury, Carlton X. Osborne, Zhandra Entrolezo, Syd Redner, Bryan Ramirez, Jonathan A. Eisen, Madeleine Leahy, Chase Keaton, Janine Wong, Jennifer Gardy, Jessica K. Jarett

**Affiliations:** 1AnimalBiome, 400 29th Street, Suite 101, Oakland, CA 94609, USA; 2Department of Evolution and Ecology, University of California, Davis, CA 95616, USA; 3Bill & Melinda Gates Foundation, Seattle, WA 98109, USA

**Keywords:** core microbiome, healthy reference, diet, age, FIV infection, gut microbiome, fecal microbiome, domestic cats

## Abstract

**Simple Summary:**

We surveyed the fecal microbial communities (termed ‘microbiome’) of North American pet domestic cats to further our understanding of the range of variation found in a population of apparently healthy cats. We also investigated whether differences in fecal microbial communities were significantly associated with the age, diet, and living environment of the individual. Results showed that thirty different bacteria were found in the fecal microbiomes of most cats. The composition of the fecal microbiome depended on the diet of the cat, their age, and whether the cat lived in a private home or a shelter. This study helped establish the expected ranges for the structure of these communities within a healthy population of cats and provides valuable insights for veterinarians, pet owners, and pet-related industries.

**Abstract:**

Here, we present a taxonomically defined fecal microbiome dataset for healthy domestic cats (*Felis catus*) fed a range of commercial diets. We used this healthy reference dataset to explore how age, diet, and living environment correlate with fecal microbiome composition. Thirty core bacterial genera were identified. *Prevotella*, *Bacteroides*, *Collinsella*, *Blautia*, and *Megasphaera* were the most abundant, and *Bacteroides*, *Blautia*, *Lachnoclostridium*, *Sutterella*, and *Ruminococcus gnavus* were the most prevalent. While community composition remained relatively stable across different age classes, the number of core taxa present decreased significantly with age. Fecal microbiome composition varied with host diet type. Cats fed kibble had a slightly, but significantly greater number of core taxa compared to cats not fed any kibble. The core microbiomes of cats fed some raw food contained taxa not as highly prevalent or abundant as cats fed diets that included kibble. Living environment also had a large effect on fecal microbiome composition. Cats living in homes differed significantly from those in shelters and had a greater portion of their microbiomes represented by core taxa. Collectively our work reinforces the findings that age, diet, and living environment are important factors to consider when defining a core microbiome in a population.

## 1. Introduction

Despite the inherent complexities of the gut microbiome and its interactions with host systems, microbiome data is advancing our collective understanding of health and disease. For example, a number of chronic diseases afflicting human populations have significantly altered gut microbiome compositions when compared to the microbiomes from healthy cohorts. These conditions include obesity, diabetes, Irritable Bowel Syndrome (IBS), periodontitis, and skin diseases, among other metabolic and immune-related conditions [1,2,3,4,5,6]. Likewise domestic cats have health conditions that are also associated with imbalances (dysbiosis) in the composition of gut bacteria, such as Inflammatory Bowel Disease (IBD), lymphoma, diabetes, periodontal disease, and atopic dermatitis [7,8,9,10,11].

Because individuals have distinctly personal microbiome signatures, aggregating microbiome data from many individuals within a natural population yields a dataset with significant compositional heterogeneity. As a result, simply characterizing the composition of a healthy microbiome is both a major challenge and a necessary first step towards identifying disease state-associated imbalances [12,13,14]. This is important for not only elucidating microbial signatures of disease states, but also understanding how the presence and/or abundance of specific taxa might be manipulated to drive the microbiome composition towards that of the disease-free state [12].

As evidenced by human studies, it is difficult to define a singular healthy host microbiome due to vast differences in the diets, lifestyles, geographic regions, and other factors that influence the composition of microbial taxa in individuals [15,16,17]. To date, studies that investigate associations between the microbiome and disease typically define ‘healthy’ as the absence of disease, but this does not necessarily mean that these individuals are ‘healthy’ [18]. Therefore, it is essential to build a comprehensive dataset of the microbiome composition of healthy individuals to establish the expected ranges for the structure of these communities within a healthy population. Such a database can then be rigorously interrogated to, for example, understand therapeutic targets. By reducing variation through a carefully defined healthy reference set, we will have more power to detect signals associated with disease states.

In this study, we aim to taxonomically define a healthy fecal microbiome dataset of North American pet domestic cats (*Felis catus*) fed a range of commercial diets. We implemented strict criteria for which samples were included in the analysis of this healthy reference dataset, with only 8.9% of samples (161 out of 1859) qualifying. To our knowledge, this is the most comprehensive and largest fecal microbiome reference set for an animal host to date [19,20]. Modeled off the American Gut Project [15], our study collected feline fecal samples through a citizen science initiative called ‘The Kittybiome Project’ [21]. We used our feline healthy reference dataset to extract and define the core microbiome of healthy domestic cats as a foundation for investigating host health and the factors that influence it.

Here, we report the core taxa and their ranges of healthy domestic cats and how age, diet, and living environment correlate with feline microbiome composition within the context of the healthy reference dataset. We also explore how a clearly diagnosed health condition, feline immunodeficiency virus (FIV) in shelter animals, compares to the taxa and ranges of the healthy reference dataset and to shelter animals without FIV. This study provides valuable insight into the ongoing discussion about the best ways to study the core microbiome and highlights the many potential applications for its use.

## 2. Materials and Methods

### 2.1. Participatory Research

In 2015, we launched a collaborative research project called “Kittybiome: kitty microbiomes for cat health and biology” to sequence fecal microbiome samples from pet domestic cats (*Felis catus*) across the country. Modeled on a similar participatory research project for humans where participants submit fecal samples for bacterial characterization [15,16,17], we sought to characterize the gut microbiome of domestic cats with a range of health conditions, ages, environments (house environment versus shelter environment), and diets [21]. Prior to launch, we received approval from the Institutional Review Board (IRB) at University of California, Davis to collect survey data for the project, and we also obtained scientific permits. Because the fecal samples are considered waste material and were going to be obtained in a non-invasive manner, the UC Davis Institutional Animal Care and Use Committee (IACUC) determined that no animal care protocol was required. This research project was later expanded to include samples collected by customers of AnimalBiome (an animal health company founded in 2016) who agreed to participate in research on the cat fecal microbiome and animal health. 

In order to understand the effect that the animal shelter environment may have on the cat fecal microbiome, we also collected samples from cats residing in the following four shelters and sanctuaries: the Berkeley Animal Care Services (Berkeley, CA, USA), Contra Costa County Animal Services Department (Martinez, CA, USA), Nine Lives Foundation (Redwood City, CA, USA), and the Regional Animal Protection Society Cat Sanctuary (Richmond, BC, Canada). We obtained 61 samples from shelter cats, and among these were 28 cats diagnosed with feline immunodeficiency virus (FIV). Samples from shelter cats where FIV status was not specifically provided were assumed to be FIV− because this is a near-universal screening test conducted at animal shelters. Samples from ten house cats with a mild health condition and five house cats diagnosed with FIV were used to test classifier models (see 3.4 Effect of FIV Status and Living Environment on the Microbiome).

### 2.2. Sample and Metadata Collection

Study participants were sent supplies needed to collect a small fecal sample that was returned by mail. Sample vials (2 mL screw cap tubes) contained 100% molecular grade ethanol and silica beads. The same materials were used to collect fecal samples from cats living in shelters, except these were collected in person. Participants were asked to complete an online survey with information about their pet cat. Fields collected in the survey included name, address, date of birth, body weight, spay or neuter status, breed, diet, clinical signs (e.g., diarrhea, vomiting, constipation), prior antibiotic exposure, diagnoses (if applicable), and any known health condition. Because a lot of information about diet and health was included in open data fields, the metadata for all samples were manually edited to include this information.

### 2.3. Sample Processing and Sequencing

All samples were processed using standardized protocols based on the Earth Microbiome Project [22]. Upon receipt, fecal material was isolated from the preservation buffer by pelleting (centrifugation at 10,000× *g* for 5 min, pouring off supernatant), and genomic DNA was extracted using the 100-prep Qiagen DNeasy PowerSoil DNA Isolation Kit (Germantown, MD, USA). Briefly, samples were placed in bead tubes containing C1 solution and incubated at 65 °C for 10 min, and this was followed by 2 min of bead beating. After, the manufacturer’s protocol was followed as written.

Amplicon libraries of the V4 region of the 16S rRNA gene (primers 505F/816R) were generated using a dual-indexing one-step PCR with complete fusion primers (Ultramers, Integrated DNA Technologies, Coralville, IA, USA) with multiple barcodes (indices) [23]. PCR reactions contained 0.3–30 ng template DNA, 0.1 μL Phusion High-Fidelity DNA Polymerase (Thermo Fisher, Waltham, MA, USA), 1X HF PCR Buffer, 0.2 mM each dNTP, and 10 μM of the forward and reverse fusion primers. The PCR conditions were as follows: initial denaturing at 98 °C for 30 s, 30 cycles of 10 s at 98 °C, 30 s at 55 °C, and 30 s at 72 °C, 30, an incubation at 72 °C for 4 min 30 s for a final extension, and a hold at 6 °C.

PCR products were assessed by running on 2% E-Gels with SYBR Safe (Thermo Fisher, Waltham, MA, USA) with the E-Gel Low Range Ladder (Thermo Fisher, Waltham, MA, USA), then purified and normalized using the SequalPrep Normalization Kit (Thermo Fisher, Waltham, MA, USA) and pooled. The final libraries were quantified with QUBIT dsDNA HS assay (Thermo Fisher, Waltham, MA, USA), diluted to 1.5 pM and denatured according to Illumina’s specifications for the MiniSeq. Identically treated phiX was included in the sequencing reaction at 25%. Paired-end sequencing of the V4 region (2 × 150 bp) was performed on the MiniSeq (Illumina, San Diego, CA, USA).

### 2.4. Sequence Data Processing

All sequence data were processed and analyzed with QIIME2 (v. 2021.8) [24]. After sequences were demultiplexed, the q2-dada2 plugin (v. 2019.1) [25] was used for quality filtering, removal of phiX, chimeric, and erroneous sequences, and identification of amplicon sequence variants (ASVs) at 100% nucleotide identity. ASV sequences were classified against the SILVA reference database (v. 132) [26,27] with the q2-feature-classifier plugin trained on the 515–806 region of the 16S rRNA gene. ASVs were collapsed based on genus level taxonomy, and samples with less than 5000 reads were removed and excluded from all analyses. When the genus was not known for a particular sequence, or if the label given by Silva was “uncultured”, “uncultured bacterium”, “uncultured organism”, or “gut metagenome”, then the label for family was used. Statistical analyses and plotting were performed in R (v. 4.1.1) [28] except where noted.

### 2.5. Core Microbiome of Healthy Cats

Of the 1859 samples collected from August 2015 through May 2021, a total of 161 samples from unique cat individuals comprised what we define as a healthy reference set (Table 1). The requirements were: a body condition score between 3 and 6 (inclusive) or a calculated BMI of less than or equal to 50, no clinical signs, no diagnoses, no antibiotics within the previous 12 months, and an age within 0.5–12 years (inclusive). This age range captures the average lifespan of cats which is 12–18 years. We did not include cats above the age of 12 years because within this cohort, these cats were reported to have health conditions (except for six individuals). Animals with missing data in these fields were also excluded. Cats receiving the following medications and supplements were excluded: probiotics, steroids, sucralfate, oclacitinib (Apoquel), cetirizine (Zyrtec), or benazepril. Cats living in shelters or sanctuaries at the time of sampling were also not included in the healthy reference due to limited information on their health histories and clinical signs.

The core fecal microbiome of healthy cats (*n* = 161) was determined by identifying genus-level bacterial taxa that were found in at least 55% of samples at a threshold of at least 25 reads per sample. We calculated the “core microbiome sum” as the summed relative abundance of all core taxa, and samples below the 2.5th percentile value for this parameter (that is, where the core taxa comprise much less of the sample than is typical) were removed from downstream analyses. From remaining samples, we calculated summary statistics and the 2.5th, 10th, 90th, and 97.5th percentile values for each taxon. We plotted distributions of taxon relative abundance and tested these distributions for unimodality with the diptest package [29]. The proportion of the core taxa that were present for each sample (e.g., number of core taxa in sample divided by the total number of core taxa) was also calculated, which we termed “core taxa present”.

### 2.6. Effect of Age on the Microbiome

We tested the effect of age on microbiome composition, alpha-diversity, and beta-diversity in healthy cats (*n* = 161). A total of six senior cats (>12 years) that were not part of the healthy reference set because of the age cutoff and otherwise met the criteria for being apparently healthy, were included in this analysis, bringing the total sample size to 167 (Table 1). Cats were categorized into the following discrete age categories: junior (7 months–3 years), prime (3–7 years), mature (7–12 years), and senior (12–14 years). Four metrics of alpha diversity (Observed genera, Shannon diversity, Gini-Simpson index, and Pielou’s Evenness) were calculated for each sample using the R microbiome package (v. 1.14.0) (for Pielou’s evenness) [30] and the R phyloseq package (v. 1.36.0) (for the other three metrics) [31].

To test whether fecal microbiome alpha-diversity correlated with host age, linear models were constructed for each alpha diversity metric and each core microbiome metric (core microbiome sum, core taxa present) using the lm function of the R stats package (v. 4.1.1) [28]. Models specified age in years as a continuous predictor variable. For beta-diversity analyses, we first calculated Bray–Curtis dissimilarity from genus relative abundances using phyloseq. Differences in microbiome composition between the discrete age categories were tested using Permutational Multivariate Analysis of Variance (PERMANOVA) on Bray–Curtis dissimilarity using the adonis2 function from the R vegan package (v. 2.6-2) [32]. The grouping of samples was visualized with principal coordinate analysis (PCoA) plots. Pairwise comparisons of microbiome beta-diversity between age categories were conducted with the EcolUtils package (v. 0.1) [33]. Differences in microbiome dispersion were tested with Permutational Multivariate Analysis of Dispersion (PERMDISP) analyses on the same data using the same package [32]. To identify the bacterial genera that were differentially abundant among age groups, we used MaAsLin2 (v. 1.7.3) [34] in R. Briefly, unrarefied relative abundance data was normalized with total sum scaling, log transformed, and all genera at a minimum prevalence of 10% and minimum relative abundance of 0.1% in all samples were analyzed with linear models that set age as a continuous predictor. The false discovery rate for multiple comparisons was set to 5% (α = 0.05).

### 2.7. Effect of Diet on the Microbiome

A total of thirty-three healthy cats had missing diet information or non-specific diet information and were excluded from all analyses involving host diet, as were two cats that had a diet of both dry and raw food (a rare combination). Thus, for diet related analyses, samples from a total of 126 healthy cats were included (*n* = 126). Information on diet was conceptualized into diet components (e.g., includes dry food (Y/N), includes wet food (Y/N), or includes raw food (Y/N) for statistical analysis, and as diet combinations (e.g., wet and dry; dry, wet, and raw) for visualization. Several diet combinations were present in only a small number of cats in the dataset, so the use of diet components for statistical analyses allowed these cats to be included for more powerful tests on the effects of diet. Interactions between diet components were not tested due to insufficient data and/or highly unequal sample numbers for some interactions.

Analyses on alpha diversity and the two core microbiome metrics were tested with linear models as described above, with the three diet components as fixed terms. If linear models were not appropriate because the values were not normally distributed, Kruskal–Wallis tests and pairwise Wilcoxon tests were used. The effect of diet components on microbiome composition or microbiome dispersion was tested with PERMANOVA and PERMDISP tests, and the differential abundance analyses were done with MaAsLin2 as described above.

Because many of the cats in the healthy reference set were fed a diet that included dry food, we explored the effect of this diet component in greater detail. We used a supervised learning approach to determine whether samples from cats that ate dry food and cats that did not could be differentiated based on their microbiome compositions. The q2-sample-classifer plugin [35] from the QIIME 2 “sample-classifier-ncv” pipeline was used to train the Random Forest classifier. Briefly, nested five-fold cross-validation was selected to ensure that all features (genus-level taxa) were tested for relative importance to the model. Two hundred estimators were used, and parameter tuning and feature selection optimization were enabled to select the optimal features and the optimal number of features for the model, with recursive feature elimination.

### 2.8. Effect of FIV Status and Living Environment on the Microbiome

To test the effect of FIV status and living environment on the microbiome, we compared the fecal microbiomes of healthy house cats (*n* = 41) with those of shelter cats with FIV (*n* = 28) and without FIV (*n* = 33). An additional five FIV+ cats living in homes were also included in the analysis. Cats from shelters were primarily in the junior (7 months–3 years) or prime (3–7 years) age categories when data on age was available. Not all shelter cats had their age recorded, but statistically these age groups comprise the majority of the population at the sampled shelters. Because most cats living in shelters are fed a combination of dry and wet food, we selected 41 cats from the healthy reference set that matched these age and diet criteria when conducting our analyses.

One Random Forest classifier was trained to distinguish FIV+ cats, FIV− shelter cats, and the age- and diet-matched subset of the healthy reference set of house cats, using the sample-classifier pipeline as described above. The only exception was using a simple five-fold cross-validation rather than a nested cross-validation. In order to more fully investigate the signal of living environment in the microbiome of healthy cats, FIV+ cats from both shelters and homes were removed from this dataset to eliminate this potentially confounding variable. Thus, the second classifier was trained to discriminate between healthy cats living in homes and FIV negative shelter cats.

Healthy house cats, FIV positive shelter cats, and FIV negative shelter cats were compared based on alpha diversity and core microbiome metrics as described above where these variables were normally distributed, as determined by Shapiro tests. If values were not normally distributed, Kruskal–Wallis and pairwise Wilcoxon rank-sum tests were used. PERMANOVA tests, PERMDISP analyses, and PCoA ordination plots were conducted as were described for the age-related analyses.

Shelter cats may be more likely than house cats to be occultly unhealthy due to more limited opportunities to observe clinical signs, unknown health histories, or newly developed health conditions. To investigate this further, we curated a test set of 10 mildly symptomatic house cats to test whether the classifier pre-trained on FIV− shelter and house cats would be able to correctly group them as “house” cats rather than “shelter” cats. The healthy mildly symptomatic cats were of junior or prime age, ate a wet and dry diet, and experienced one or two of the following clinical signs: diarrhea, vomiting, constipation, or compulsive grooming.

## 3. Results

### 3.1. The Core Microbiome of Healthy Pet Cats

Samples from a total of 161 healthy cats living in households were used to describe the core fecal microbiome in healthy pet cats (Table 1). Thirty genus-level taxa from five phyla were determined to be part of the “core” microbiome in cats (Table 2). The core genera with the highest relative abundances were *Prevotella*, *Bacteroides*, *Collinsella*, *Blautia*, and *Megasphaera*. The most prevalent core genera were *Bacteroides*, *Blautia*, *Lachnoclostridium*, *Sutterella*, and *Ruminococcus gnavus*. Most distributions of taxon relative abundances were markedly positively skewed, with a long tail of higher values (Appendix A). A few genera showed possible multimodal distributions (e.g., *Prevotella*, *Ruminococcus gauvreauii*), although no statistically significant deviations from unimodality were detected for any of the genera tested (Appendix A).

### 3.2. Effect of Age on the Microbiome

Bacterial community composition remained relatively stable across the different age classes of healthy pet cats (Figure 1A). The most abundant bacterial genera across samples were *Prevotella*, *Bacteroides*, and *Fusobacterium*. The total percentage of the fecal microbiome comprised by the core taxa did not change with age (Figure 1B, PERMANOVA F = 0.26, R^2^ = 0.002, *p* > 0.05).

However, the number of core taxa that were present (characterized as a % of the 30 taxa), decreased significantly with age (Figure 1B, PERMANOVA F = 8.364, R^2^ = 0.048, *p* < 0.01). Thus, while cats lose members of the core taxa as they age, it appears that populations of other core members of the community may increase in relative abundance as a result.

Furthermore, no significant differences in microbiome alpha diversity were observed among cats of different ages (Figure 1C, Appendix A), and age class did not significantly predict fecal microbiome beta-diversity either (Figure 1D, PERMANOVA F = 1.56, R^2^ = 0.027, *p* = 0.08). We did identify three genera that were differentially abundant among age groups. Overall, the relative abundances of *Fusicatenibacter*, *Subdoligranulum*, and *Megasphaera* decreased as the host aged (Appendix A). *Subdoligranulum* and *Megasphaera* were highly prevalent among healthy cats and were found in >62% (100/161) of individuals from the healthy reference set. The opposite was true of *Fusicatenibacter*, which had a low prevalence in the health reference set, and was detected in only 14% (24/161) of cats in the healthy reference.

### 3.3. Effects of Diet on the Microbiome

Of the various diets fed to the set of healthy (primarily North American) cats, the most common combination was canned wet food with dry kibble (Appendix A). Overall, dry food was a component of the diet for 72% of healthy cats in the dataset for which diet information was known (Appendix A). Fecal microbiome composition varied with host diet category in healthy cats (Figure 2A); cats that ate raw food for example, contained larger relative abundances of *Collinsella* in their gut microbiomes than the rest of the surveyed cats (Figure 2A). Cats that ate purely dry food appeared to have lower relative abundances of *Peptoclostridium* compared to cats with other diets.

The total percentage of the fecal microbiome made up by the core taxa tended to be slightly higher in cats that included dry kibble in their diet than cats that did not include dry kibble in their diet (Figure 2B, Appendix A). Furthermore, individuals that ate raw food had fewer core taxa in their fecal microbiomes compared to individuals that did not have any raw food in their diet (Figure 2B, Appendix A), but the combined relative abundance of the fecal microbiome composed by the core taxa did not vary between the two groups. Thus, the core microbiome of raw-fed cats appears to contain taxa that are not highly prevalent or as abundant in the gut microbiomes of other individuals. No differences in core microbiome metrics were observed between cats that ate wet food and those that did not (Figure 2B, Appendix A).

When examining fecal microbiome alpha-diversity, differences among dietary types were detected. Generally, fecal microbiome richness and evenness were lowest among cats that incorporated dry food in their diet compared to the rest of the surveyed individuals (Figure 2C, Appendix A). Fecal microbiome evenness (Shannon diversity, Simpson’s index, and Pielou’s evenness) was highest among cats whose diet contained wet food over cats whose diet did not contain wet food (Figure 2C, Appendix A). Furthermore, fecal microbiome beta-diversity also varied with host diet, and the largest difference was found between cats that included dry food in their diet compared to cats that did not (Figure 2D, Appendix A). Microbiome dispersion, which is a measure of community variability or heterogeneity in a group of samples, varied with diet type, specifically, between cats that ate wet food and those that did not eat any wet food (Appendix A). Thus, differences detected for this variable could also be partially attributed to variability in fecal microbiome dispersions.

According to differential abundance analyses, 21 bacterial genera were significantly associated with dry food, 2 with raw food, and 14 with wet food (Appendix A); 14 of the 21 genera were members of the core microbiome. *Prevotella*, *Megamonas*, and *Megasphaera* for example, were found at larger relative abundances in the fecal microbiomes of cats that ate dry food vs. cats that did not (Appendix A). *Fusobacterium* relative abundances were lower in dry food fed cats than the other surveyed cats. The bacterial genera *Blautia* and *Faecalibacterium* were enriched in fecal microbiomes of cats that ate some wet food over those that did not (Appendix A). Lastly, cats with raw food in their diet tended to harbor fewer *Faecalibacterium* than other cats. A total of five bacterial taxa were associated with more than one diet type (diet component): *Ruminococcus torques*, *Tyzzerella*, *Faecalibacterium*, *Subdoligranulum*, and *Faecalitalea* (Appendix A).

In addition to differential abundance analyses, we also used a supervised machine learning approach to determine whether samples from the different diets could be differentiated based on their microbiome compositions. The random forest classifier was highly accurate (96%) at distinguishing cats that consumed any dry food from those that did not. A total of 24 bacterial genera comprised half of the model’s importance, and 9 of these taxa were members of the core microbiome (Appendix A). *Alloprevotella* relative abundances appeared to be enriched in the fecal microbiomes of cats that consumed dry food, while the opposite was true of *Romboutsia* relative abundances (Appendix A). Interestingly, half of these taxa (12/24) were also singled out by our differential abundance analyses as being significantly associated with cats that ate dry food (Appendix A). These results suggest that a dry food diet is a strong predictor of fecal microbiome composition, and that the gut microbiomes of cats that eat any amount of dry food is distinct from those of cats that do not eat any dry food.

### 3.4. Effects of FIV Status and Living Environment on the Microbiome

Finally, we investigated whether the fecal microbiomes of domestic cats varied with host living environment (house vs. shelter), and feline immunodeficiency virus (FIV) status (negative vs. positive) (Appendix A). For this, we conducted analyses comparing the fecal microbiomes of healthy house cats, FIV negative shelter cats, and FIV positive shelter cats. When examining fecal microbiome composition, both groups of shelter cats harbored lower relative abundances of *Bacteroides*, *Blautia*, *Collinsella*, and *Peptoclostridium* compared to healthy house cats (Figure 3A). Cats diagnosed with FIV had similar microbiome compositions as cats without FIV (Figure 3A). Additionally, compared to healthy house cats, both groups of shelter cats had fewer core microbiome genera and a smaller percentage of their microbiome represented by core genera (Figure 3B, Appendix A). Cats with FIV had the same number of core genera and the same percentage of the microbiome represented by core taxa as did cats negative for FIV (Figure 3B, Appendix A).

Healthy house cats and both groups of shelter cats had fecal microbiomes of equal richness, but healthy cats had more even fecal microbiomes than shelter cats (Figure 3C, Appendix A). Shelter cats diagnosed with FIV had fecal microbiomes of similar richness and evenness as shelter cats without FIV (Figure 3C, Appendix A). Overall, it appears living environment is a strong predictor of fecal microbiome composition, while differences relating to FIV status are only apparent when comparing FIV positive shelter cats with healthy house cats. If cats come from the same living environment, the potential influences of FIV status are diminished.

Similar to diet analyses, we used a random forest classifier to determine whether healthy house cats had distinct fecal microbiome signatures from those of FIV− and FIV+ shelter cats. The classifier did not perform well at discriminating between the two groups of shelter cats (FIV+ vs. FIV−). For FIV+ cats, 66% of the test set was correctly assigned, and for FIV− cats, 43% of the set was correctly assigned. However, the classifier performed well at distinguishing the healthy reference set from both groups of shelter cats to 100% accuracy. A total of 27 bacterial genera comprised half of that model’s importance, and 8 of these taxa were members of the core microbiome (Appendix A). Specifically, both groups of shelter cats harbored greater relative abundances of *Lactobacillus*, *Lactococcus*, *Peptococcus*, and unclassified *Prevotellaceae* than cats living in homes (Appendix A). Conversely, home cats were enriched in *Bacteroides*, *Peptoclostridium*, and *Collinsella* compared to shelter cats (Appendix A). Among shelter cats, cats negative for FIV had slightly more *Campylobacter* and uncultured *Lachnospiraceae* than cats with FIV, which instead were enriched in *Ruminococcus gnavus* (Appendix A).

Lastly, we tested whether there was a bias in our random forest model by determining whether shelter cats were easily distinguishable from house cats because they were occultly unhealthy. For this, we compared the gut microbiomes of mildly symptomatic house cats with FIV negative shelter cats. The model achieved an overall accuracy of 80% and only two out of ten house cats were incorrectly classified as shelter cats. The model’s performance decreased relative to the previous test set, but was still above random chance. These results suggest that the gut microbiomes of shelter cats and indoor cats are distinct and this is not due to differences in their health condition.

## 4. Discussion

If we are to use insights gained from microbiome testing to develop new diagnostics and therapeutics, it is essential to develop reference sets from populations of healthy individuals. In this study, we explored a dataset of 16S rRNA gene sequence data derived from fecal samples collected from >1800 pet cats, focusing on the <10% (161 samples) reported to have been collected from apparently healthy cats. We used this healthy reference dataset to explore how age, diet, feline immunodeficiency virus (FIV) status, and living environment correlate with fecal microbiome composition, alpha-diversity, and beta-diversity. In this population of healthy cats, we found that host diet and living environment were more influential factors than age and FIV status.

### 4.1. Core Microbiome

The ranges of 30 core genera found in at least 55% of individuals in a population of apparently healthy cats were statistically defined, with *Prevotella*, *Bacteroides*, *Collinsella*, *Catenibacterium*, *Blautia*, *Faecalibacterium*, and *Megasphaera* among the highest in relative abundance in the feline fecal microbiome. These results are consistent with prior studies that found that *Bacteroides* and *Prevotella* are main bacterial constituents of the healthy feline gut microbiota [36]. Even when using culture-dependent methods, isolates classified as *Bacteroides* are the most frequently recovered from cat fecal samples [37]. Prior studies found that *Faecalibacterium* are also common in the fecal microbiomes of healthy kittens fed a range of diets [38].

Many of the core genera detected in cat fecal microbiomes are associated with the synthesis of short-chain fatty acids (SCFAs; e.g., acetate, butyrate, and propionate) and branched-chain fatty acids (BCFAs; e.g., valerate, isobutyrate, and isovalerate) from the fermentation of carbohydrates, protein, and fiber in the GI tract [39,40,41]. Both SCFAs and BCFAs act as indispensable energy sources for host colonocytes, stimulate colonic blood flow and motility and promote the growth of commensal and resident bacteria, outcompeting pathogenic microbes [42,43,44]. Additionally, SCFAs have been linked to appetite suppression and weight loss in rodents and humans [45]. There is increasing evidence that gut microbial metabolites may act as regulators of gene expression via their interactions with histone deacetylases [46]. They may also act as signaling molecules via their agonistic interactions with inflammatory G protein-coupled receptors [46]. This body of work highlights the wide systemic effects that gut bacteria and their metabolites have on the host and its associated microbes (the holobiont).

*Prevotella*, *Catenibacterium,* and *Megasphaera* are highly correlated with propionate production in the mammalian intestine and have the metabolic machinery to produce propionate from the digestion of glucose or lactate [40]. *Faecalibacterium* are strongly correlated with acetate concentrations, can ferment dietary carbohydrates into SCFAs like acetate, and can further convert this acetate to butyrate [47]. This is significant as the SCFAs differ in their potential impacts on host physiology. Butyrate is preferentially used by the gut mucosa, propionate contributes to gluconeogenesis in the liver and acetate is most concentrated in the blood [47]. Thus, the proportions of core bacterial genera and their metabolic byproducts could have widespread effects on host functioning.

### 4.2. Age

The fecal microbiome composition was fairly consistent among the different age classes in healthy pet cats. In terms of alpha and beta diversity measures, younger cats did not have vastly different fecal microbiomes from those of mature or senior cats. However, the total number of core taxa in these age classes declined significantly with age, even though the percentage of the microbiome comprised by core taxa did not change.

Prior studies report that in kittens, the gut microbiome undergoes a period of change at 18 to 30 weeks and then stabilizes after 42 weeks, with this change being primarily attributed to dietary transitions [48]. At 18 weeks of age, the kitten fecal microbiome is dominated by the genera *Lactobacillus* (35% average relative abundance) and *Bifidobacterium* (20%) whereas at 42 weeks of age the genera with the highest relative abundances are *Bacteroides* (16%), *Prevotella* (14%) and *Megasphaera* (8%) [48]. In our study, we did not focus on the development of the fecal microbiome in kittens and did not include cats under seven months of age, with most over a year old. Thus, the window where differences in the microbiome associated with weaning and early life had likely already passed.

A number of studies have identified changes in intestinal function associated with aging in a number of mammalian species, including a reduction in microbiome alpha-diversity, and slower intestinal transit times [49]. Here, we focused on individuals with no known health conditions, which becomes increasingly challenging with age. It is possible that some of the studies reporting age associated changes in the microbiome in individuals are actually associated with health conditions that become more prevalent with age.

A previously published study reported age related changes in the intestinal microbiota of pre-weaned, young, mature, and geriatric cats using culture-based methods and real time PCR [50]. They reported that *Eubacterium* were more frequently isolated from pre-weaned cats while *Bifidobacterium* from geriatric cats, and *Lactobacillus* from younger cats [50]. However, these findings come from culture-based studies and are not directly comparable to composition-wide surveys of the gut microbiome that are achieved from next-generation sequencing.

### 4.3. Diet

In our study, the fecal microbiome composition of healthy cats varied with diet. The fecal microbiomes of cats fed any amount of dry food were significantly different from those of cats not fed any dry kibble. Our random forest classifier was highly accurate at distinguishing cats that ate dry food from cats that did not eat dry food and achieved 96% accuracy. It is clear that there is something fundamentally different about the fecal microbiomes of cats that ingest any type of dry food. Furthermore, fecal microbiomes of cats with raw food included in their diet were enriched in different bacterial genera than cats that were not fed any raw food. This is not surprising as dry kibble, commercial wet food, and raw-based diets contain different ratios of protein, carbohydrates, and fat, which promote the growth of different bacteria and may lead to distinct gut microbiome compositions [51,52,53]. Raw food diets for example, are high in protein and low in fiber, whereas wet food diets have slightly less protein and more fat [51,52,53]. Kibble diets have slightly less protein and fat than canned wet food, and much more carbohydrates than canned wet food or raw food [51,52,53]. Bacteria will vary in their abilities to metabolize, ferment, or transform those dietary components or their derivatives, leading to the enrichment of different microbes under different diets. Raw food diet in particular may enrich for different microbes because of its high protein intake, given that this food usually consists of pure skeletal muscle, fat, organs, cartilage, and bones from small animals [54].

In our study, cats fed dry kibble harbored greater relative abundances of *Prevotella* and *Megasphaera* compared to cats that did not eat any dry kibble. In a previously published study, cats fed dry kibble diets also tended to be enriched in *Prevotella* compared to cats fed raw food; those cats were instead enriched in *Clostridium* and *Fusobacterium* [40]. This makes sense as *Prevotella* are prominent carbohydrate utilizers and are abundant in the fecal microbiomes of cats fed moderate protein diets [55], whereas *Clostridium* typically degrade protein and are associated with high-protein diets in cats and dogs [56]. Similarly, a study found differences in the gut microbiomes of cats fed a moderate protein and high carbohydrate diet (32% protein, 32% carbohydrates) and cats fed a high protein, low carbohydrate diet (51% protein, 11% carbohydrates) [53]. Specifically, cats consuming less protein had larger abundances of *Megasphaera* whereas those fed more protein had larger abundances of *Faecalibacterium*. Lastly, a third study compared the microbiomes of cats fed dry kibble vs. commercial wet food and found that cats fed dry kibble harbored greater relative abundances of *Megasphaera* and *Lactobacillus*, and less *Blautia* than cats fed wet food [52].

### 4.4. Environment

In our study, our random classifier was highly accurate at distinguishing healthy house cats from shelter cats (positive or negative for FIV) based purely on their fecal microbiome compositions. The microbiomes of healthy house cats were distinct from those of shelter cats and there are several reasons as to why this might be the case. Compared to house cats, shelter cats might experience more crowded housing, less play time and human interactions, and lesser quality diets [57,58]. Their shelters might be noisier than private homes and may also house dogs. Shelter cats may be handled more than typical indoor house cats, or may be more likely to experience changing caregivers, all of which could increase their stress and potentially impact their microbiome [58,59]. Conversely, indoor house cats might also receive more quality time with their owner, and their owner might have more resources to maximize the cat’s welfare and wellbeing, potentially shaping their microbiome. It has also been shown that there might be greater sharing and exchange of microbes between cat owners and their pets, thus, indoor house cats might have slightly different compositions than shelter animals due to their close physical contact with their human owners [60].

Furthermore, cats in shelters are also more likely to be exposed to or infected with pathogens. In a study examining the incidences of infection among house cats, shelter cats, and stray cats, shelter cats were two times more likely to be infected by at least one enteric parasite (e.g., *Toxocara cati*, *Cytospora* spp., *Giardia* spp., *Dipylidiidae* or *Taeniidae* tapeworms, *Isospora* spp., etc.) than cats in private homes [61,62]. Shelter cats might also be more likely to contract a respiratory disease because of the crowded housing conditions [63]. It is widely known that the gut microbiomes of wild and domestic mammals with a range of infections are distinct from those of healthy animals [64,65]. Regardless of whether the shelter cats in our study were or were not currently diagnosed with infections, the conditions of their living environment could be contributing to their divergent microbiomes.

### 4.5. FIV Status

While the fecal microbiome compositions of shelter cats with FIV were not fundamentally distinct from those without FIV, they were distinct from the fecal microbiomes of healthy house cats. This is understandable as FIV infections cause a weakening of the immune system and are a source of significant morbidity and mortality in cats [66,67]. Cats diagnosed with FIV may have a greater risk of developing lymphoma, neurological dysfunction, and secondary opportunistic infections due to their immunocompromised state [66,67]. This retroviral infection understandably impacts a cat’s health and physiology, which may lead to consequential changes in the microbiome. Similar to our study, a prior study also found that the fecal microbiome of cats with an FIV infection differed from those of uninfected controls [68]. Authors stated that FIV-infected cats had greater relative abundances of *Acidobacteria* and *Actinobacteria* and fewer *Fibrobacteres* in their microbiomes than control cats [68]. Our findings also echo prior microbiome studies which report associations between the gut microbiome and other viral infections including astrovirus (AstV) infections in Jamaican fruit bats [69], Acquired Immune Deficiency Syndrome (AIDS) in chimps [70], and Adenovirus (AdV) infections in Malagasy mouse lemurs [71].

## 5. Limitations

This study employed 16S rRNA amplicon sequencing which provides information on the composition of the fecal microbiome, but at a limited taxonomic resolution. Future studies will benefit from the use of whole-genome sequencing, shotgun sequencing, or other -omics technologies to determine the specific bacterial species and bacterial functions that are present in the fecal microbiomes of healthy pet domestic cats. Moreover, our study used broad dietary categories (dry food, wet food, raw food) to examine the relationship between host diet and the microbiome. To get a more in-depth understanding of what exactly may be driving differences in the microbiome, we need to directly measure the macronutrient and nutritional content of the cat diets.

The study provided valuable insights regarding the diversity of microbiome compositions present in healthy domestic cats of a single species. It would be interesting to examine whether these findings also apply to populations of wild cats from other species (e.g., leopards, pumas, or tigers).

Lastly, the study did not examine sex- and breed-related differences in the fecal microbiomes of healthy pet cats, but a permutational ANOVA indicated that these influences were not significant in this dataset. The fecal microbiomes of cats in this study did not vary by sex (marginal PERMANOVA R^2^ = 0.072, *p* = 0.39) or breed (marginal PERMANOVA R^2^ = 0.008, *p* = 0.12).

## 6. Conclusions

In this study, we built and examined a healthy reference dataset composed of microbiome samples collected from healthy pet cats living in homes. Additional samples were also collected from cats living in shelters or sanctuaries and compared with the healthy reference set. While some effects of age and FIV status on fecal microbiome composition were detected, host diet and living environment were more influential factors, and should be carefully considered when designing future studies. Thus, we show that the ‘healthy’ cat fecal microbiome does not look one certain or averaged way, but is rather represented by a diversity of compositions that are dependent on host characteristics and lifestyle. Our analyses helped establish the expected ranges for the structure of these communities within a healthy population of cats, which will be useful to veterinarians, pet owners, and pet-related industries.

## Figures and Tables

**Figure 1 vetsci-09-00635-f001:**
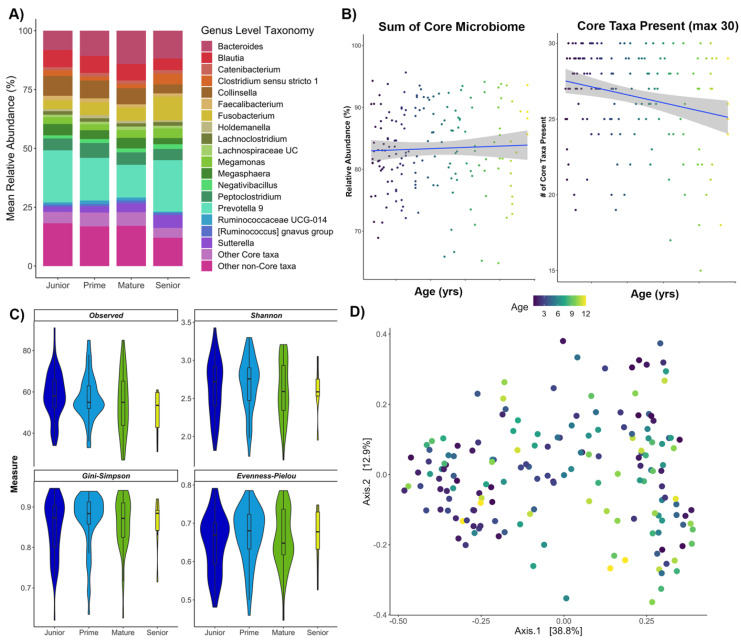
The fecal microbiomes of cats across different ages. (**A**) Stacked bar plot depicting the relative abundances of the most abundant bacterial genera in fecal samples by age class. When the genus was not known, the label for Family was used. (**B**) Scatter of the relationship between two core microbiome metrics (percentage of microbiome comprised by core; total number of core taxa present) and age (years) as a continuous variable. (**C**) Violin plots of microbiome alpha-diversity (Observed Richness, Shannon diversity, Gini-Simpson index, and Pielou’s Evenness) by age class. (**D**) PCoA ordination based on Bray–Curtis dissimilarity. Closeness of points indicates microbiome similarity and points are color-coded by host age (years).

**Figure 2 vetsci-09-00635-f002:**
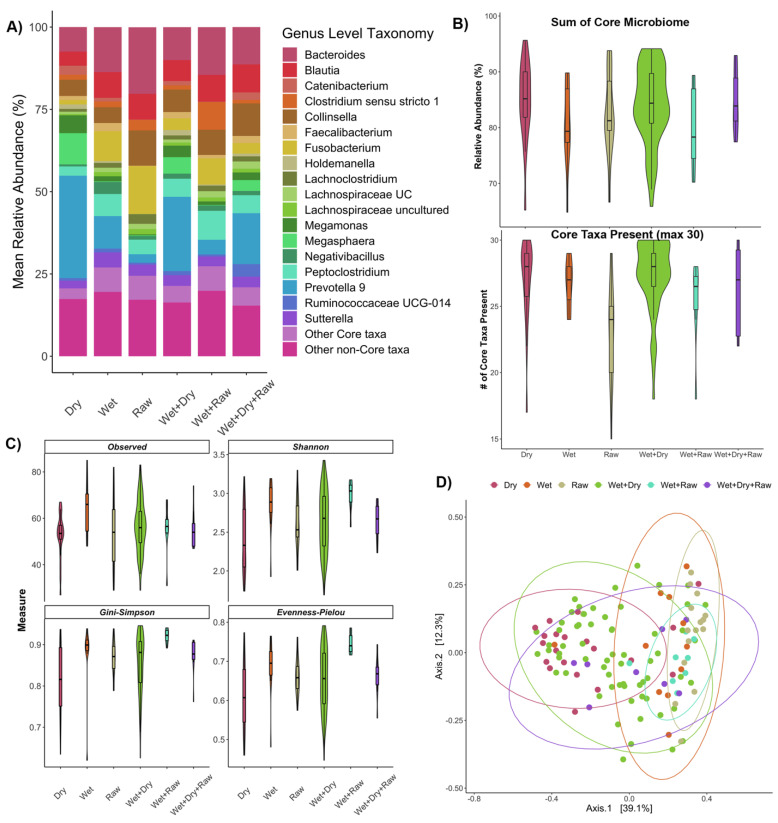
Fecal microbiome variation is significantly associated with host diet category. (**A**) Stacked bar plot depicting the relative abundances of the most abundant bacterial genera in fecal samples. Samples are categorized by diet category. (**B**) Violin plots showing the values of two core microbiome metrics (percentage of microbiome comprised by core; total number of core taxa present) by diet category. (**C**) Violin plots of microbiome alpha-diversity (Observed Richness, Shannon diversity, Gini-Simpson index, and Pielou’s Evenness) by diet category. (**D**) PCoA ordination based on Bray–Curtis dissimilarity. Closeness of points indicates microbiome similarity and points are color-coded by diet category.

**Figure 3 vetsci-09-00635-f003:**
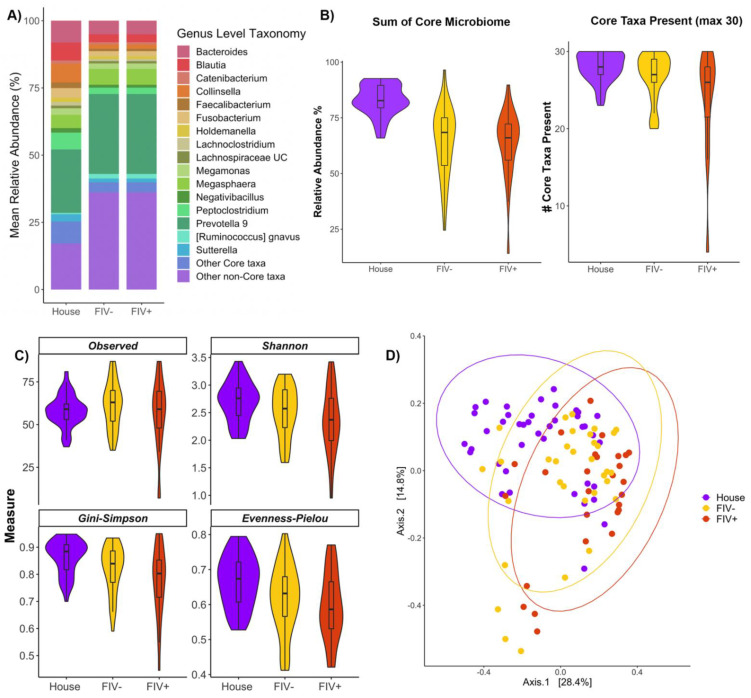
The fecal microbiome is distinct between indoor house cats and shelter cats. (**A**) Stacked bar plot depicting the relative abundances of the most abundant bacterial genera in fecal samples. Samples are categorized by host environment (house vs. shelter) and FIV status (negative vs. positive). Note that both FIV+ and FIV− cats came from shelters. (**B**) Violin plots showing the values of two core microbiome metrics (percentage of microbiome comprised by core; total number of core taxa present) for house cats and FIV+ or − shelter cats. (**C**) Violin plots of microbiome alpha-diversity (Observed Richness, Shannon diversity, Gini-Simpson index, and Pielou’s Evenness) by FIV status and environment type. (**D**) PCoA ordination based on Bray–Curtis dissimilarity. Closeness of points indicates microbiome similarity and points are color-coded by FIV status and environment type.

**Table 1 vetsci-09-00635-t001:** Characteristics of cats included in the healthy reference set (*n* = 161), and six senior cats that were only included in age-related analyses (*n* = 6).

Age Class	*n*	Age (Months)	Fecal Score	Body Condition Score	F (%)	SN (%)
Junior	58	21.5 (±8.48)	2.7 (±1.04)	5.2 (±0.38)	47	95
Prime	59	58.7 (±15.28)	2.7 (±0.95)	5 (±0.61)	61	86
Mature	44	109.5 (±15.84)	3.1 (±0.99)	5.1 (±0.56)	59	93
Senior	6	141.3 (±4.32)	3.2 (±0.5)	5.6 (±0.55)	50	83

Note: Fecal scores were reported for 116 cats and body condition scores were reported for 145 cats. Abbreviations: BCS = Body Condition Score, F (%) = percentage of age class that were female, SN (%) = percentage of age class reported to be spayed or neutered.

**Table 2 vetsci-09-00635-t002:** Thirty bacterial genera comprise the core microbiome in healthy pet cats.

Phylum	Class	Order	Family	Genus	Median	SD	Prev.	2.5th	10th	90th	97.5th
Actinobacteria	Coriobacteriia	Coriobacteriales	Coriobacteriaceae	Collinsella	5.93	6.456	96.4	0.81	1.62	16.98	25.35
Actinobacteria	Coriobacteriia	Coriobacteriales	Eggerthellaceae	Slackia	0.15	0.178	55.6	0.04	0.06	0.51	0.66
Bacteroidetes	Bacteroidia	Bacteroidales	Bacteroidaceae	Bacteroides	8.34	9.421	100	0.45	1.3	25.74	32.25
Bacteroidetes	Bacteroidia	Bacteroidales	Prevotellaceae	Prevotella 9	23.17	17.55	80.5	0.11	1.02	46.51	56.69
Bacteroidetes	Bacteroidia	Bacteroidales	Tannerellaceae	Parabacteroides	0.51	1.038	80.5	0.08	0.14	1.57	3.71
Firmicutes	Clostridia	Clostridiales	Clostridiaceae	Clostridium sensu stricto 1	1.2	3.237	84	0.06	0.15	6.39	11.39
Firmicutes	Clostridia	Clostridiales	Lachnospiraceae	Ruminococcus gauvreauii	0.37	0.503	90.5	0.07	0.13	1.05	1.67
Firmicutes	Clostridia	Clostridiales	Lachnospiraceae	Ruminococcus gnavus	0.51	1.478	97	0.06	0.11	2.4	5.1
Firmicutes	Clostridia	Clostridiales	Lachnospiraceae	Ruminococcus torques	0.6	0.818	91.1	0.08	0.16	2.12	2.8
Firmicutes	Clostridia	Clostridiales	Lachnospiraceae	Blautia	5.44	5.65	100	1	1.84	13.88	20.96
Firmicutes	Clostridia	Clostridiales	Lachnospiraceae	Lachnoclostridium	1.13	1.103	98.2	0.2	0.4	3	4.28
Firmicutes	Clostridia	Clostridiales	Lachnospiraceae	Lachnospiraceae NK4A136	0.28	0.43	86.4	0.05	0.11	0.83	1.68
Firmicutes	Clostridia	Clostridiales	Lachnospiraceae	Unclassified Lachnospiraceae	0.92	1.165	96.4	0.13	0.28	2.63	3.72
Firmicutes	Clostridia	Clostridiales	Lachnospiraceae	Lachnospiraceae UCG-009	0.18	0.223	72.2	0.05	0.07	0.57	0.94
Firmicutes	Clostridia	Clostridiales	Lachnospiraceae	Lachnospiraceae uncultured	0.8	0.888	93.5	0.11	0.21	2.04	3.22
Firmicutes	Clostridia	Clostridiales	Peptococcaceae	Peptococcus	0.57	1.37	65.1	0.09	0.13	2.64	4.62
Firmicutes	Clostridia	Clostridiales	Peptostreptococcaceae	Peptoclostridium	4.52	5.047	89.3	0.55	1.19	12.82	19.28
Firmicutes	Clostridia	Clostridiales	Ruminococcaceae	Butyricicoccus	0.17	0.28	68	0.04	0.07	0.45	0.92
Firmicutes	Clostridia	Clostridiales	Ruminococcaceae	Faecalibacterium	1.38	1.95	79.9	0.08	0.24	4.72	7.49
Firmicutes	Clostridia	Clostridiales	Ruminococcaceae	Negativibacillus	1.19	1.529	95.3	0.11	0.23	3.58	5.69
Firmicutes	Clostridia	Clostridiales	Ruminococcaceae	Oscillibacter	0.25	0.327	79.9	0.04	0.08	0.61	0.99
Firmicutes	Clostridia	Clostridiales	Ruminococcaceae	Ruminiclostridium 9	0.43	0.458	84.6	0.05	0.08	1.15	1.65
Firmicutes	Clostridia	Clostridiales	Ruminococcaceae	Ruminococcaceae UCG-014	0.61	3.432	74.6	0.06	0.12	3.07	9.41
Firmicutes	Clostridia	Clostridiales	Ruminococcaceae	Ruminococcaceae uncultured	0.15	0.255	60.4	0.03	0.05	0.48	0.95
Firmicutes	Erysipelotrichia	Erysipelotrichales	Erysipelotrichaceae	Catenibacterium	2.01	2.179	56.2	0.16	0.39	5.6	7.42
Firmicutes	Erysipelotrichia	Erysipelotrichales	Erysipelotrichaceae	Holdemanella	0.85	2.371	66.9	0.08	0.17	3.32	10.01
Firmicutes	Negativicutes	Selenomonadales	Veillonellaceae	Megamonas	1.81	5.355	75.7	0.1	0.15	8.37	19.25
Firmicutes	Negativicutes	Selenomonadales	Veillonellaceae	Megasphaera	5.21	7.796	58	0.15	0.4	16.67	28.39
Fusobacteria	Fusobacteriia	Fusobacteriales	Fusobacteriaceae	Fusobacterium	3.95	6.149	88.8	0.11	0.28	13.43	21.86
Proteo bacteria	Gamma proteobacteria	Betaproteobacteriales	Burkholderiaceae	Sutterella	2.23	2.846	97	0.14	0.46	6.28	9.85
Core Microbiome Total					82.59	6.775	NA	67.65	74.07	91.59	93.99

Shown are the taxonomic designations (Silva v. 132) of thirty bacterial taxa identified as being part of the core microbiome in healthy cats. These were all bacterial genera that were found in at least 55% of samples at a threshold of at least 25 reads per sample. Also shown are the median, standard deviation, prevalence, and summary statistics for the 2.5th, 10th, 90th, and 97.5th percentile values for each taxon. Not all sequences were classified to genera and in those instances, their last known classification (e.g., Family) was used.

## Data Availability

Raw 16S rRNA gene sequence files, ASV tables, sample metadata, QIIME2 code, and R code are available upon request for academic use. These data are not publicly available due to their commercial value.

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
