# Peer review of "The Kitty Microbiome Project: Defining the Healthy Fecal “Core Microbiome” in Pet Domestic Cats"

_vetsci, 2022, doi:10.3390/vetsci9110635_

Round 1

Reviewer 1 Report

The authors investigated fecal samples from cats with different ages, diets and locations and FIV conditions to develop a healthy reference microbiome database for domesticated cats. 

While the "Kittybiome" project is interesting to understand feline biomes, the findings from the current work such as microbiome changes with diet, age and environment or infectious disease are well known and can offer limited information. Although the authors have put in their best efforts in collating such a massive dataset- this can be used model or reference and may not be a complete picture. 16s sequencing is well-known and is an upgrade from culture-bases method, however, as the technologies is emerging an evolving further, smWGS or similar may provide us the complete microbiome up to species-level. 

Still there is a huge debate about what is healthy or Healthy vs normal microbiome, this is an exciting area of research and has a lot to do in times come. For that reason, the current work will be helpful and steps towards understanding the healthy microbiome. 

Abstract

Lines: 29-30, This is well-known concept and the word "Healthy individuals" looks out of context? 

Methods

Little to know information about the health characteristics of the cats and whether they are sex-matched. 

Avg lifespan of cats needs to mentioned. 

Majorly, the authors have not mentioned about cat species, while research clearly indicates inter individual differences, population and sex-based differences, cat species-species difference will be an important aspect to consider.

Results

Whether are any differences with sexes as seen in other mammals? It will be a wonderful opportunity to add this to reference microbiome. 

At least in domesticated cats, if there is a way to track their health metrics, such as their vaccine status, which may have an impact on their biomes? 

Author Response

We thank the reviewer for taking the time to provide their valuable feedback. We are pleased to see that they thought our introduction provided sufficient background and that our results were clearly presented. Per the reviewer’s recommendations, we have added missing details to the methods section regarding the sampling of healthy cats.

The authors investigated fecal samples from cats with different ages, diets and locations and FIV conditions to develop a healthy reference microbiome database for domesticated cats. 

While the "Kittybiome" project is interesting to understand feline biomes, the findings from the current work such as microbiome changes with diet, age and environment or infectious disease are well known and can offer limited information. Although the authors have put in their best efforts in collating such a massive dataset- this can be used model or reference and may not be a complete picture. 16s sequencing is well-known and is an upgrade from culture-bases method, however, as the technologies is emerging an evolving further, smWGS or similar may provide us the complete microbiome up to species-level. 

Still there is a huge debate about what is healthy or Healthy vs normal microbiome, this is an exciting area of research and has a lot to do in times come. For that reason, the current work will be helpful and steps towards understanding the healthy microbiome.

We thank the reviewer for their honest feedback. We agree that our findings are not necessarily novel and echo previously published work. What makes our work unique is that we explicitly established the expected ranges for the structure of fecal microbiomes within a healthy population of cats, which had not been done before. 

We agree that whole-genome sequencing can provide additional insights regarding microbiome composition (to species-level) and function and encourage future studies to use WGS and additional -omics to get an even more detailed pictures of the microbiome. We added these details to our Limitations section in the Discussion (L677-681):

“This study employed 16S rRNA amplicon sequencing which provides information on bacterial composition, but at a limited taxonomic resolution. Future studies will benefit from the use of whole-genome sequencing, shotgun sequencing, or other -omics technologies to determine the specific bacterial species and bacterial functions that are present in the fecal microbiomes of healthy pet domestic cats.” 

Abstract

Lines: 29-30, This is well-known concept and the word "Healthy individuals" looks out of context? 

We agree with the reviewer that our work provides further evidence to support well-known findings that microbiome variation is related to host diet, age and the environment. We have modified our sentence to reflect this (and have removed the words ‘healthy individuals’) (L75-77):

“Collectively our work reinforces the findings that age, diet, and living environment are important factors to consider when defining a core microbiome in a population.”

Methods

Little to know information about the health characteristics of the cats and whether they are sex-matched. 

Information on the health characteristics of sampled cats can be found under the ‘Core microbiome of healthy cats’ section in the Methods. We describe in detail the age, sex, BMI, and health conditions of the sampled individuals (L211-222):

“Of the 1,859 samples collected from August 2015 through May 2021, a total of 161 samples from unique cat individuals comprised what we define as a healthy reference set (Table 1). The requirements were: a body condition score between 3 and 6 (inclusive) or a calculated BMI of less than or equal to 50, no clinical signs, no diagnoses, no antibiotics within the previous 12 months, and an age within 0.5-12 years (inclusive). This age range captures the average lifespan of cats which is 12-18 years. We did not include cats above the age of 12 years because within this cohort, these cats were reported to have health conditions (except for six individuals). Animals with missing data in these fields were also excluded. Cats receiving the following medications and supplements were excluded: probiotics, steroids, sucralfate, oclacitinib (Apoquel), cetirizine (Zyrtec), or benazepril. Cats living in shelters or sanctuaries at the time of sampling were also not included in the healthy reference due to limited information on their health histories and clinical signs.” 

Approximately 50-60% of fecal samples from healthy pet cats were female (Table 1), so the sex ratio is more or less even.

If there are additional details the reviewer wants us to include, please let us know what these are and we would be happy to include them.

Avg lifespan of cats needs to mentioned. 

We have added a sentence to the methods stating the average lifespan of cats (L215-216):

“This age range captured the average lifespan of cats, which is 12-18 years”

Majorly, the authors have not mentioned about cat species, while research clearly indicates inter individual differences, population and sex-based differences, cat species-species difference will be an important aspect to consider.

We agree with the reviewer that microbiome composition can be species-specific and correlated with host phylogeny. In our paper we analyzed the fecal microbiomes of a single cat species (Felis catus) but we hope future studies conduct similar analyses on other cat species and in wild cats to see how trends compare and contrast. We added this recommendation to our Limitations section: (L687-690):

“The study provided valuable insights regarding the diversity of microbiome compositions present in healthy domestic cats of a single species. It would be interesting to examine whether these findings also apply to populations of wild cats from other species (e.g., leopards, pumas, or tigers).”

Breed is another important factor to consider. In dogs, breed differences in the microbiome may be confounded by body size (Deschamps et al. 2022), which is less of an influence on cats since they do not exhibit as much variation in body size. In cats, the skin and oral microbiomes of hairless Sphynx cats are distinct from those of Bengal cats (Felis catus x Prionailurus bengalensis) (Older et al. 2019). 

We tested whether there was an effect of breed on the microbiome in healthy pet cats using a permutational ANOVA and found it was not significant (R2=0.00887, p=0.12). Most of the cats in this population were mixed breed domestic short, medium, and long haired cats.

Deschamps C, Humbert D, Zentek J, Denis S, Priymenko N, Apper E, Blanquet-Diot S. From Chihuahua to Saint-Bernard: how did digestion and microbiota evolve with dog sizes. Int J Biol Sci. 2022 Aug 1;18(13):5086-5102. doi: 10.7150/ijbs.72770. PMID: 35982892; PMCID: PMC9379419.

Older, C. E., Diesel, A. B., Lawhon, S. D., Queiroz, C. R., Henker, L. C., & Rodrigues Hoffmann, A. (2019). The feline cutaneous and oral microbiota are influenced by breed and environment. PloS one, 14(7), e0220463.

Results

Whether are any differences with sexes as seen in other mammals? It will be a wonderful opportunity to add this to reference microbiome. 

We did not analyze sex differences in the microbiome as previous research indicated that these differences are likely to be minimal in cats (Deusch et al. 2015), and if such differences are present we would expect spayed females and neutered males to be very similar, as is the case in dogs (Scarsella et al. 2020). This information in combination with the high rate of spaying and neutering in the cat population studied here, led us to believe that investigating sex differences would not be of interest. However, in response to the reviewer’s suggestion, we ran a permutational ANOVA to test for differences in sex and found it was not significant (R2=0.072, p=0.39), and added these findings to the manuscript (L692-696):

“Lastly, the study did not examine sex- and breed-related differences in the fecal microbiomes of healthy pet cats, but a permutational ANOVA indicated that these influences were not significant in this dataset. The fecal microbiomes of cats in this study did not vary by sex (marginal PERMANOVA R2= 0.072, p=0.39) or breed (marginal PERMANOVA R2= 0.008, p=0.12).” 

Deusch, O., O’Flynn, C., Colyer, A., Swanson, K. S., Allaway, D., & Morris, P. (2015). A longitudinal study of the feline faecal microbiome identifies changes into early adulthood irrespective of sexual development. PloS one, 10(12), e0144881.

Scarsella, E., Stefanon, B., Cintio, M., Licastro, D., Sgorlon, S., Dal Monego, S., & Sandri, M. (2020). Learning machine approach reveals microbial signatures of diet and sex in dog. PLoS One, 15(8), e0237874.

At least in domesticated cats, if there is a way to track their health metrics, such as their vaccine status, which may have an impact on their biomes? 

The reviewer makes an important point that the cat’s health may influence the microbiome. For our study, we set very strict criteria to minimize the influences of confounding medical conditions on the microbiome. Our criteria were a body condition score between 3 and 6 (inclusive) or a calculated BMI of less than or equal to 50, no clinical signs, no illnesses/disorders/diagnoses, no antibiotics within the previous 12 months, and an age within 0.5-12 years (inclusive). 

In this study, we did not explore whether vaccination affects the core healthy microbiome in pet cats because the majority of cats participating in the citizen science project were adopted from North American animal shelters where vaccination is obligatory.

Reviewer 2 Report

Suggest that the authors move discussion of the inclusion/exclusion criteria to lines 100-102 or wait to elaborate on the # of samples collected/analyzed until this is discussed later (line 173)

Were sample collection procedures the same for shelter cats as for the remainder of the subjects?

Fig 1b, Fib 2B have overlapping labels

Are the authors able to provide any additional analysis of wet vs dry food (e.g., macronutrients) that could be correlated w/ the microbiome, or is there too much variation in the brand/type of food within those overarching categories of wet/dry? This would be very interesting!

Overall this is a well presented and thoroughly investigated study. The writing is clear and concise, and the data is presented well. 

Author Response

We thank the reviewer for their thoughtful feedback. We are pleased to see that the reviewer (based on their responses to questions) found our manuscript well-written, our results clearly presented, and our conclusions adequately supported. Per their recommendations, we included a few missing details to our methods section.

Suggest that the authors move discussion of the inclusion/exclusion criteria to lines 100-102 or wait to elaborate on the # of samples collected/analyzed until this is discussed later (line 173)

Per the reviewer’s recommendations, we moved the sentences detailing the number of samples collected from the ‘Participatory research’ paragraph to the ‘Core microbiome of healthy cats’ paragraph (L211-214):

“Of the 1,859 samples collected from August 2015 through May 2021, a total of 161 samples from unique cat individuals comprised what we define as a healthy reference set (Table 1). The requirements were: a body condition score between 3 and 6 (inclusive) or a calculated BMI of less than or equal to 50, no clinical signs [...]”

Were sample collection procedures the same for shelter cats as for the remainder of the subjects?

The procedures for collecting fecal samples for shelter cats were the same as they were for collecting samples from healthy cats (2 ml screw cap tubes containing 100% molecular grade ethanol and silica beads). We have added a sentence to the methods section to clarify (L159-162):

“Study participants were sent supplies needed to collect a small fecal sample that was returned by mail. Sample vials (2 ml screw cap tubes) contained 100% molecular grade ethanol and silica beads. The same materials were used to collect fecal samples from cats living in shelters, except these were collected in person.”

Fig 1b, Fib 2B have overlapping labels

We thank the reviewer for pointing this out. During the upload process, some of the figure labels were slightly altered/blurred but this issue has been fixed in the revised manuscript.

Are the authors able to provide any additional analysis of wet vs dry food (e.g., macronutrients) that could be correlated w/ the microbiome, or is there too much variation in the brand/type of food within those overarching categories of wet/dry? This would be very interesting!

We agree with the reviewer that these analyses would be interesting indeed! Because our study relied on citizen science and obtaining fecal samples given voluntarily by the public, we could not readily measure the macronutrients in the food consumed by the pet cats. With hundreds of flavors, varieties and brands of cat food available, it is also difficult to obtain sufficient detail from citizen science participants to reference the manufacturer’s published guaranteed analysis.The impact of different diets on the microbiome is a huge focus of the nutrition research being conducted in companion animals right now. Several reviews have been written summarizing this research, including one that we do cite in the Discussion section:

Wernimont, Susan M., et al. "The effects of nutrition on the gastrointestinal microbiome of cats and dogs: impact on health and disease." Frontiers in Microbiology 11 (2020): 1266.

We state in our Limitations section that directly measuring the nutritional content of the cat diets would be beneficial (L681-685):

“Moreover, our study used broad dietary categories (dry food, wet food, raw food) to examine the relationship between host diet and the microbiome. To get a more in-depth understanding of what exactly may be driving differences in the microbiome, we need to directly measure the macronutrient and nutritional content of the cat diets.”

Overall this is a well presented and thoroughly investigated study. The writing is clear and concise, and the data is presented well. 

Thank you! We appreciate the reviewer’s comments.

Round 2

Reviewer 1 Report

No further comments